# Circulating MicroRNAs as Novel Biomarkers for Osteoporosis and Fragility Fracture Risk: Is There a Use in Assessment Risk?

**DOI:** 10.3390/ijms21186927

**Published:** 2020-09-21

**Authors:** Simone Ciuffi, Simone Donati, Francesca Marini, Gaia Palmini, Ettore Luzi, Maria Luisa Brandi

**Affiliations:** 1Department of Experimental and Clinical Biomedical Sciences, University of Florence, Viale Pieraccini 6, 50139 Florence, Italy; simone.ciuffi@unifi.it (S.C.); simone.donati11@gmail.com (S.D.); f.marini@dmi.unifi.it (F.M.); gaia.palmini@unifi.it (G.P.); ettore.luzi@unifi.it (E.L.); 2Unit of Bone and Mineral Diseases, University Hospital of Florence, Largo Palagi 1, 50139 Florence, Italy; 3Fondazione Italiana Ricerca sulle Malattie dell’Osso (FIRMO Onlus), 50141 Florence, Italy

**Keywords:** osteoporosis, fragility fractures, circulating microRNAs, biofluids biopsies, diagnostic biomarkers

## Abstract

Osteoporosis is a multifactorial skeletal disease that is associated with both bone mass decline and microstructure damage. The fragility fractures—especially those affecting the femur—that embody the clinical manifestation of this pathology continue to be a great medical and socioeconomic challenge worldwide. The currently available diagnostic tools, such as dual energy X-ray absorptiometry, Fracture Risk Assessment Tool (FRAX) score, and bone turnover markers, show limited specificity and sensitivity; therefore, the identification of alternative approaches is necessary. As a result of their advantageous features, such as non-invasiveness, biofluid stability, and easy detection, circulating cell-free miRs are promising new potential biomarkers for the diagnosis of osteoporosis and low-traumatic fracture risk assessment. However, due to the absence of both standardized pre-analytical, analytical, and post-analytical protocols for their measurement and universally accepted guidelines for diagnostic use, their clinical utility is limited. The aim of this review was to record all the data currently available in the literature concerning the use of circulating microRNAs as both potential biomarkers for osteoporosis diagnosis and fragility fracture risk evaluation, and group them according to the experimental designs, in order to support a more conscious choice of miRs for future research in this field.

## 1. Introduction

Osteoporosis (OP) is a systemic skeletal disorder, which is common in older people, particularly in postmenopausal (PM) women, and it is characterized by increased fragility fracture risk due to bone mass reduction and the deterioration of bone tissue microarchitecture [1]. In fact, osteoporosis diagnosis is often made after low-traumatic fracture occurrence. At the cellular level, OP is triggered by a progressive imbalance between bone-forming osteoblast cell (OB) and bone-resorbing osteoclast cell (OC) activity, tending toward the latter [2]. According to the World Health Organization, OP is one of the 10 most common diseases, affecting approximately 200 million women worldwide [3]. Hip fractures cause chronic pain symptoms, and they are associated with reduced mobility, disability, and a loss of autonomy more than all other types of fracture [4], resulting in increased morbidity, with reported mortality rates up to 20–24% in the first year after the fracture event [5]. It has been estimated that 2.7 million hip fractures occurred worldwide in 2010 [4], and that this number is expected to increase to 6.3 million in 2050 [3]. In Europe, hip fractures account for 67% of the total health-care expenditure for OP, for a total of approximately €24 billion, and these costs are expected to double by the year 2050 [3]. These significant forecasts clearly highlight the importance of accurate and effective tools for both OP diagnosis and fragility fracture risk assessment in order to reduce costs of osteoporotic patient management. Currently, the gold standard methodology is represented by dual energy X-ray absorptiometry (DXA), which is used to measure bone mineral density (BMD) [6]. Another validated tool that is often used in clinical routine along with DXA, is the Fracture Risk Assessment Tool (FRAX) score, which calculates the 10-year probability of a major (FRAX-M) and hip osteoporotic fracture (FRAX-H) [6]. In addition, bone turnover markers (BTMs), such as C-terminal type I collagen crosslinks (CTx), type I pro-collagen pro-peptide (PINP), pyridinoline/deoxypyridinoline, parathyroid hormone (PTH), osteocalcin, bone alkaline phosphatase (BAP), and tartrate-resistant acid phosphatase 5b (TRAP5b), can be used to evaluate bone cell metabolic activity and the efficiency of the anti-resorptive therapies [7]. However, to date, these diagnostic tools present practical defects: DXA can give information only if an established architectural alteration was already made, which requires weeks or months to detect [8]; FRAX score excludes some important variables, such as an increased risk of subsequent fractures after the initial fracture [9]; BTMs are not completely specific in evaluating bone formation and resorption [7]. Consequently, although these diagnostic tools are valuable, new alternative biomarkers are desirable, which alone or combined with the previous ones allow greater efficacy in both OP diagnosis and fragility fracture risk assessment. In this context, circulating cell-free microRNAs (c-miRs) have gained increased attention as potential OP diagnostic biomarkers during recent years [10]. MicroRNAs (miRs) are endogenous short, single-stranded non-coding RNAs (18–25 nucleotides in length) that reversibly inhibit gene expression via binding to the 3′ untranslated region of the mRNA target [11]. Since their discovery in *Caenorhabditis elegans* in 1993 by Lee and colleagues [12], miRs have been identified in all kingdoms [13], and their crucial regulatory role in various physiological and pathological processes, including bone metabolism, has been well established [14]. In 2008, four independent studies identified extracellular miRs in blood and showed that the changes in their levels can be indicative of both physiological and pathological conditions [15,16,17,18]. To date, these small molecules can be isolated from a variety of biofluids including plasma, serum, urine, and saliva [19]. Moreover, their elevated stability in biological fluids has been shown, and it is imputable to encapsulation in membrane-bound vesicles, such as shedding microvesicles, exosomes, and apoptotic bodies [20], or to their association with proteins, such as Argonaute 2, high-density lipoproteins, and nucleophosmin 1 [21]. Finally, these molecules possess the three fundamental features to be used as biomarkers. (1) They are easily obtainable (with minimal or non-invasive procedures) and detectable in a reproducible manner with high specificity and sensitivity (via various platforms: real-time qRT-PCR, microarray, and NGS). (2) They can add new information with regard to disease of interest (3) They help the physicians’ decision-making [22]. These features make c-miRs candidates for novel potential diagnostic biomarkers. Indeed, starting in 2014, many studies aiming to identify specific c-miRs as potential diagnostic biomarkers for OP have been conducted. Nevertheless, the promising data obtained from these studies are not homogeneous [10], which is probably due to an absence of standardized and validated pre-analytical (i.e., source/matrix type), analytical (i.e., quantification platform used [23]), and post-analytical (i.e., reference genes selected [24]) protocols, and to the differences in study design. These variables have limited the use of c-miRs as diagnostic biomarkers in clinical practice to date.

## 2. Aim

The purpose of this review is to try to summarize the results obtained on the use of c-miRs as biomarkers for OP diagnosis and fragility fracture risk assessment, and group them based on experimental design similarity, with the belief that this can contribute to the minimization of the above variables, in order to support a more conscious choice of miRs for future research in this field.

For this purpose, after careful PubMed literature research, we have identified three main clinical categories of comparison: (1) OP or osteopenia (Ope) without fragility fracture (woF) patients vs. normal bone mass controls (CTRs) woF volunteers; (2) OP or Ope with fragility fracture (wF) vs. CTRs woF; (3) OP or Ope wF vs. CTRs with non-OP fractures (wnOF). In addition, although most studies only recruited postmenopausal (PM) women, some also evaluated c-miRs in premenopausal (PRM) women and young men with idiopathic osteoporosis, so we decided to create subgroups that also took these characteristics into account.

## 3. C-miRs in OP

### 3.1. C-miR Signatures in OP/Ope woF

#### 3.1.1. PM OP/Ope vs. PM CTRs

The first study aiming to identify c-miR signatures in PM OP and Ope woF was performed in 2014 by Li H et al. [25]. According to their T-scores (total hip BMD), 120 Chinese women were grouped in 40 OP, 40 Ope, and 40 age-matched PM woF CTRs. Plasma levels of three miRs that are known to be involved in osteoporosis, such as miR-21, miR-133a, and miR-146a, were evaluated by real-time qRT-PCR (from here on called qPCR) platform. Expression data normalization was performed by using miR-16 as an endogenous reference gene (eRG). They found a significant miR-133a up-regulation and miR-21 down-regulation in OP and Ope patients vs. the PM CTRs group. Moreover, miR-21 plasma levels showed positive correlations and miR-133a plasma levels showed negative correlation with both hip and spine BMD in all three subcohorts, concluding that both of these miRs could be used to accurately discriminate OP and Ope patients from healthy woF CTRs.

By using microarray methodology, Chen et al. [26] analyzed ovariectomy (OVX) rat serum levels of 758 miRs as a discovery stage, identifying 35 dysregulated miRs with respect to the female rat sham-operated control group. Fourteen of these miRs, and the miR-328-3p selected from the literature, were validated as dysregulated in the same rat model by qPCR, using miR-25-3p as the eRG. Starting from these data, human serum levels of these 15 miRs were assessed in 10 PM OP and 7 PM Ope woF, who were compared to 19 healthy woF PM women, also by qPCR. They found that miR-328a-3p, miR-103-3p, and miR-142-3p were down-regulated in PM OP, while miR-30b-5p was up-regulated in both PM OP and Ope. Moreover, each of these four miRs shows a positive correlation with hip BMD, and, in addition, miR-30b-5p and miR-142-3p were also positively correlated with femoral neck BMD. Finally, ROC analysis revealed their potential as diagnostic biomarkers for distinguishing OP/Ope from CTRs.

A cohort of elderly Slovenian PM woF women were divided into 17 OP and 57 CTRs, based on femoral neck, total hip, and lumbar spine T-score, and the plasma levels of nine miRs previously found as dysregulated in bone tissue from OP patients by the authors [27] were investigated. By using qPCR platform and a combination of let-7a-5p and miR-16-5p as eRG, miR-148a-3p circulating levels were found to be significantly higher in OP patients compared to healthy women, proposing it as a potential plasma-based biomarker for OP. Furthermore, although no significant differences were found in the plasma levels between the two subgroups, a positive correlation between miR-126-3p levels and BMD of 1/3 radius, and negative correlation between miR-423-5p and FRAX-M were found.

In 2018, Liu H et al. [28] identified miR-96 as a potential biomarker for PM woF OP. Microarray analysis on serum samples from OPs (*n* = 5) and CTRs (*n* = 5) identified two miRs, miR-96 and miR-107 as up-regulated. A validation step, performed by qPCR and *U6* as eRG, confirmed that the levels of these c-miRs in elderly OP women (*n* = 20) were higher than in elderly healthy women (*n* = 20). Based on the fold change (FC) value, they chose miR-96 to evaluate its role in osteogenic differentiation. They found that miR-96 was overexpressed in human bone marrow-derived mesenchymal stem cells (hBMSCs) obtained from aged patients, when compared with hBMSCs from young patients. Moreover, they show that various osteogenic markers were down-regulated in hBMSCs transfected with agomiR-96, and the *Osterix* gene was the putative target of this miR in hBMSCs. Lastly, miR-96 serum levels were negatively correlated with bone alkaline phosphatase (sBALP) in PM woF OP. Overall, these results demonstrate the role of miR-96 in bone metabolism and as a candidate valid biomarker for OP diagnosis.

In the same years, Chen R et al. [29] compared c-miR levels in serum samples from nine elderly OP woF vs. nine elderly woF CTRs, age- and sex-matched. They initially screened 150 miRs by qPCR methodology and using *U6* as the eRG, and they identified 14 significantly aberrant expressed miRs in OP patients compared to CTRs, of which four (miR-10b-3p and miR-100 up-regulated, and miR-328-3p and let-7g-5p down-regulated) showed tight association with the Wnt pathway, and these were selected for subsequent analysis. All four identified c-miRs showed an Area Under the Curve (AUC) value greater than 0.85, as well as their role in bone metabolism by regulating ALP activity and calcium deposition in human OB lines, inducing the authors to conclude that these miRs have potential for OP diagnosis.

The role of serum miR-203 in distinguishing OP patients from normal BMD subjects, and its impact on the osteogenic differentiation of BMSCs, was evaluated by two research groups [30,31]. They recruited a cohort of 120 (60 OP and 60 CTRs) and 100 (60 OP and 40 CTRs) age-, weight-, and height-matched Chinese volunteer PM woF women, respectively. They found that the levels of miR-203 in OP women were significantly decreased in comparison with normal subjects. Moreover, by using a similar experimental protocol on BMSCs from rats and humans, respectively, they showed that the overexpression of miR-203 increased many osteogenic markers by directly targeting *DKK1* mRNA, concluding that this miR can promote BMSC differentiation in OBs.

Perksanusak and co-authors investigated miR-21 plasma levels in a cohort of 109 PM low BMD (LBMD) (14 OP and 95 Ope) woF patients and 41 PM CTRs, since it is involved in the control of bone remodeling and because its circulating levels were not influenced by dietary intake, timing of blood collection, and sex of patients [32]. This miR was found up-regulated in LBMD, and it was negatively correlated with both lumbar spine and total hip BMD. Finally, positive correlation was also found with PINP value only in OP subgroups.

Chen et al. also evaluated a panel of eight specific c-miRs in PM LBMD woF patients (*n* = 46) by a qPCR platform and using a combination of three eRGs to normalize data expression [33]. Although none of the serum levels of these miRs were significantly dysregulated, miR-21-5p and miR-23a-3p showed log_2_ FC of >1 (biologically considered up-regulated) and miR-125b-5p of <−1 (biologically considered down-regulated) in LBMD women compared to CTRs. Moreover, miR-21-5p levels were negatively correlated to left and right trochanter bone mineral content (BMC); miR-23a-3p was found to be positively correlated to TRAP5 levels; and miR-133a-3p showed a significantly positive correlation with total body BMC.

The levels of four miRs (miR-17, miR-20a, miR-21, and miR-29a) were found decreased in the serum of 48 OP patients with respect to 48 normal subjects [34]. Three of these miR (miR-20a, miR-21 and miR-29a) were also shown down-regulated in the bone tissue of the same cohort of OP patients. Based on both the lower *p*-value and higher FC, miR-21 was chosen for further investigation. The BMSCs from OP rats showed a down-expression of miR-21; moreover, various osteogenic-related markers were found up-regulated when mimic-miR-21 was transfected in rat BMSCs. Therefore, the authors conclude that miR-21 may serve as a potential therapeutic target for OP.

The studies reported below were mainly focused on the possible role of specific c-miRs, which were identified as dysregulated in OP, as targets for clinical treatment of this pathology. However, they also provide useful information regarding their potential use as a biomarker for the diagnosis of osteoporosis.

miR-579-3p was indicated as highly expressed in the serum of OP patients and able to inhibit hBMSC osteogenic differentiation [35]. In fact, the overexpression of this miR in hBMSCs induces a significant decrease of both specific osteogenic mRNA and protein expression, as well as mineralized nodule formation. Finally, in silico and function studies indicated that miR-579-3p could inhibit the osteogenic differentiation of hBMSCs via the targeting of the *Sirt1* gene, which is known to play an important role in this process. These results suggest that miR-579-3p could promote the development of OP, and consequently, it could be a valid therapeutic target for the treatment of this pathology.

Similarly to a previous study, based on both lower *p*-value and higher FC, Li et al. identified miR-373 as the best c-miR to distinguish OP patients from healthy normal BMD subjects [36]. In a cohort of 20 PM OP woF patients, serum levels of three miRs (miR-28, miR-101, and miR-373) were significantly decreased when compared to the same number of CTRs by using qPCR platform and *U6* as the eRG. Subsequent studies on both human bone tissues and rat serum, BMSCs, and bone tissues confirmed that miR-373 was down-regulated in OP vs. CTR samples; moreover, they also demonstrated its involvement in OP progression by inducing osteogenic-related markers’ down-regulation.

Interestingly, Zang H et al. [37] evaluated the serum levels of long non-coding RNA (lncRNA) XIXT and its target (miR-30a-5p) in both OP patients and CTR subjects. qPCR-based expression analysis of XIXT and miR-30a-5 showed significant down- and up-regulation in the serum of OP patients, respectively. Further studies on hBMSCs showed that alterations in the balance of cellular levels between these two molecules could cause the onset and progression of OP.

A group of Chinese researchers showed that miR-410 and its gene target (*BMP-2*) were overexpressed and down-expressed in both human PM OP and OVX mouse CD^14+^ cells, respectively, proving their potential involvement in pathological processes of OP. Moreover, they demonstrated that serum levels of miR-410 were increased in human PM OP (*n* = 26) when compared with age-matched CTRs (*n* = 29). This result was confirmed on a mouse OVX model [38].

In a similar study presented by Tang et al. [39], the role of miR-144 in the proliferation and osteogenic differentiation on rat BMSCs by targeting *SFRP1* gene was shown. Furthermore, the human serum levels, analyzed by qPCR, of this c-miR were found increased in PM OP patients (*n* = 15) with respect to PM CTRs (*n* = 15).

In a cohort of 60 voluntary donors, equally divided between OP and CTRs, Lv R and co-authors showed that miR-200a-3p was potentially involved in osteogenic mesenchymal stem cell differentiation by binding the *GLS* gene. At the same time, they showed that compared with healthy controls, serum levels of this miR were significantly higher in OP patients [40].

Lin et al. conducted a very complex study to demonstrate that miR-338 clusters, including miR-338 and miR-3065-5p, inhibited osteogenic differentiation by osteogenic markers down-regulation [41]. Simulating an in vitro paracrine model by using continuous mouse bone cell lines, they showed that an miR-338 cluster was cross-transferred between OP-engaged and healthy OBs, reducing their bone deposition activity. On the one hand, they found that the miR-338 cluster was up-regulated in the serum of PM OP (*n* = 15) with respect to age-matched PM CTRs (*n* = 15), and, on the other hand, that ROC analysis associates its levels with the OP phenotype. Interestingly, this result was not only confirmed in OVX rat serum, but it was also shown that the miR-338 cluster serum levels were significantly increased at 1 and 8 weeks post-OVX operation—therefore, much earlier than bone loss was detectable (12 weeks). These results suggest the miR-338 cluster as a potential diagnostic biomarker for OP.

miR-19a-3p is dysregulated in a variety of cancers and participates in the mediation of many biological processes, but its role in OP is not yet well known. In their study, Chen R et al. highlighted the ability of this miR to promote hBMSC osteogenic differentiation via binding the *HDAC4* gene, which was proven to inhibit osteogenic differentiation-related markers expression [42]. In addition, they reported that miR-19a-3p serum levels in OP patients (*n* = 42) were significantly lower than in healthy subjects (*n* = 42).

In a cohort of 120 PM Indonesian women, equally grouped into OP and CTR groups, serum levels of a panel of five miRs, correlated to bone quality and quantity, were evaluated by using qPCR methodology [43]. Expression analysis revealed that miR-21 was up-regulated in PM OP groups; furthermore, its levels correlated positively with serum levels of both RANKL and OPG, and negatively with BMD. Lastly, by multiple linear regression analysis, they established a cut-off point for miR-21 (30.1-copies/µL × 1010) to distinguish patients with and without OP.

Similarly, Ding W et al. showed that plasma levels of miR-100 were significantly increased in OP patient groups (*n* = 120) vs. healthy individual CTR groups (*n* = 120). Moreover, its levels were negatively correlated with BMD, lumbar spine L2-L4 T-score, and serum levels of 25OH-D3. These results, together with those obtained by ROC analysis, indicated miR-100 as a sensitive biomarker to identify OP patients [44].

Lei et al. conducted a study aimed to investigate if teriparatide administration contributed to hBMSC osteogenic differentiation by miR-375 regulation [45]. In this study, they also evaluated the serum levels of this miR in 30 OP patients compared with 30 CTR subjects. qPCR analysis revealed that miR-375 was up-regulated in OP groups. Subsequently, functional experiments showed that this miR inhibits hBMSC osteogenic via binding *RUNX2* mRNA. Conversely, teriparatide enhanced hBMSC mineralization capacity through the miR-375/RUNX2 axis.

In three recent studies, the role of miR-429 [46], miR-1286 [47], and miR-217 [48] in hBMSC osteogenic differentiation, by binding their specific mRNA target (*SCD-1*, *FZD4*, and *RUNX2* gene respectively) and using similar protocols, was functionally demonstrated. In addition, the authors reported that the serum levels of these miRs were up-regulated in OP woF patients, when compared to normal BMD individuals, by using very similar experimental protocols.

A recently published paper reported that serum levels of three miRs (miR-208a-3p, miR-155-5p, and miR-637) selected from the literature as involved in bone cell differentiation were all increased in well-characterized woF PM OP with respect to woF PM CTRs. Moreover, the levels of these circulating molecules were significantly and negatively correlated with the BMD and T-score of the lumbar spine, hip, and forearm. Finally, the diagnostic power of each of these three miRs to distinguish OP patients from healthy individuals was demonstrated by a high AUC value [49].

The next four studies were conducted on patient cohorts that included all the clinical categories considered in this review. In this section, we will report only the results obtained from PM OP/Ope and PM CTR comparison. The remaining data will be described in the following sections.

Yavropoulou et al. investigated levels of twelve bone-related miRs and two miRs reported as dysregulated in monocytes of OP patients. The serum samples were obtained from a cohort of LBMD PM patients (35 wF and 35 woF) and PM CTRs (30). By using qPCR assay, and *SNORD95*, *SNORD96A* and *RNU6-2* genes as eRG, the authors identified two up-regulated miRs (miR-124-3p and miR-2861) and three down-regulated (miR-21-5p, miR-23a, and miR-29a-3p) in patients when compared to CTRs. Moreover, miR-21-5p was also found to be down-regulated when its serum levels in fractured patients were compared with OP woF. These results suggest that these five c-miRs could be employed as biomarkers to distinguish LBMD patients from CTR subjects [50].

A total of 370 miRs were tested in two pooled serum samples from nine OP patients, eight Ope patients, and four healthy subjects by qPCR array systems. The next validation stage on a larger cohort identified two miRs, miR-122-5p and miR-4516, whose levels were decreased in the serum and plasma of patient groups, respectively. Positive correlation was found between these miRs and T-score lumbar spine. ROC analysis revealed that the best AUC value was reached by combining these two miRs. In silico studies identified many bone-related genes as potential targets of these miRs [51]. In conclusion, miR-122-5p and miR-4516 may be useful diagnostic biomarkers for OP.

The serum levels of 754 miRs were evaluated in four pooled samples each of five PM OP patients and another four prepared from five PM healthy women. Of these, seven miRs were increased in the serum of patients vs. CTRs, and three (miR-23-3p, miR-140-3p, and miR-885) were confirmed as up-regulated in the validation stage. Specifically, miR-23-3p and miR-140-3p were higher in both PM OP (*n* = 16) and Ope (*n* = 28) woF patients, compared to CTR women (*n* = 22). In addition, the serum levels of both miR-23-3p and miR-140-3p showed a negative correlation with BMD, and ROC analysis highlighted their ability to discriminate OP patients from healthy [52].

By utilizing the miR microarray dataset on the plasma of six CTRs, six OP woF and six OP wF patients, nine miRs were found to be decreased in the low bone mass group with respect to CTRs, but only the down-regulation of miR-19b-3p was confirmed in a wider cohort (24 OP woF and 24 CTRs). In addition, plasma levels of this miR positively correlated with BMD, and it exhibited an AUC value higher than 0.9. Furthermore, in vitro and in vivo studies evaluated the functional role of miR-19b-3p, proving its involvement in bone loss prevention by targeting the *PTEN* gene. Together, these findings suggested miR-19b-3p as a valid biomarker for OP and its therapeutic potential for the treatment of this pathology [53].

#### 3.1.2. PM OP vs. PRM CTRs

At the time of the PubMed medline, only four studies had evaluated c-miR signatures in PM OP patients by comparing them with PRM CTR subjects. However, one of these [54] did not identify any significant variation between the two groups in the serum levels of 33 selected miRs.

In 2016, You et al. were the first authors to compare c-miR levels in these two categories [55]. First, they performed a discovery stage on serum samples of five women with PM OP and five healthy PRM, by microarray platform, which covers 851 human miRs, identifying 33 down-regulated miRs in PM OP. One of them, miR-27a, was the most strongly down-regulated, and it is known to be involved in both osteogenic and adipogenic differentiation; therefore, it was chosen for the next validation phase. Its serum levels were singularly assessed in a cohort of 155 individuals grouped into 81 PM OP patients and 74 PRM CTRs, confirming its down-regulation in the patient group.

Liu H et al., whose work has already been partially described in the previous paragraph [28], further evaluated the serum levels of miR-96 in a cohort consisting of 20 PM OP and 20 PRM CTRs. Unexpectedly, the levels of this miR were found to be reduced in the OP group, whereas its negative correlation with BAP was confirmed, further demonstrating its potentiality as a biomarker for OP diagnosis.

Similarly, part of the Ismail scientific report has also been described above [49]. In this section, we report the results obtained on serum levels of selected miRs when PRM OP patients were compared with PMR CTR groups. They found that miR-208a-3p and miR-155-5p were up- and down-regulated in patients, respectively. Furthermore, ROC analysis confirmed their properties as biomarkers for OP.

The serum levels of three miRs were found to be significantly dysregulated in LBMD woF patients (17 OP and 14 Ope), when compared to CTRs (14 PRM normal BMD), by using the qPCR platform and *5S* as the eRG. In particular, miR-204-3p was up-regulated in both OP and Ope patients, while miR-181c-5p and miR-497-5p were down-regulated. Moreover, miR-204-3p was positively correlated with BMI and CTx and negatively correlated with BMD, whereas miR-181c-5p and miR-497-5p results were negatively correlated with the latter parameter. Lastly, ROC analysis and functional studies suggested that all three miRs may be used as biomarkers to distinguish LBMD patients from healthy subjects, and that both miR-181c-5p and miR-497-5p were involved in OB differentiation processes [56].

We close this chapter by reporting the results of a meta-analysis carried out by Pala E et al. [57]. The inclusion criteria established for the selection of studies considered only articles (*n* = 16) that evaluated c-miR levels in PM woF OP vs. PM woF CTRs. Although this meta-analysis included scientific papers published until October 2018, by the application of various specific algorithms, the up-regulated miR-133a-3p, among the 75 miRs reported to be differentially expressed in a group of 327 PM OP vs. 328 healthy CTRs, was identified as a statistically significant meta signature. Consequently, the authors state that miR-133a-3p could be used as a potential non-invasive biomarker for OP diagnosis.

Table 1 summarizes c-miRs identified as potential biomarkers for distinguishing OP/Ope woF patients from normal BMD subjects, indicating the protocols used in each study.

### 3.2. C-miR Signatures in OP wF

#### 3.2.1. OP wF vs. CTR woF

Weilner et al. [58] were the first to study the characteristic c-miR expression profile of PM wF patients, who were diagnosed with OP if they had sustained at last one low-traumatic fracture. Preliminary discovery analysis was carried out on serum samples of seven OP and seven age-matched CTRs by using qPCR-based panels that covered 175 human miRs. Of the six miRs found dysregulated, only three (miR-22-3p, miR-328-3p, and let-7g-5p) were confirmed down-regulated in a larger validation set of samples. These data convinced the authors that these three miRs could be potential candidates for both OP diagnosis and fragility fracture risk assessment.

After the pre-screening of 179 human miRs in two pooled serum sample from eight PM OP wF patients and five woF PM women with osteoarthritis, used as CTRs, and a subsequent validation step on serum of 12 CTRs and 15 patients, the levels of three miRs (miR-21-5p, miR-122-5p, and miR-125b-5p) were found increased in the latter group, although only miR-21-5p was significantly up-regulated after age adjustment. Moreover, considering the positive correlation of the serum levels with some BTMs and high AUC values, these three miRs were proposed as potential biomarkers for this pathology [59].

In order to discriminate between OP bone fracture and diabetes bone fracture, Heilmeier and co-authors evaluated the serum levels of 375 miRs in a cohort of 80 PM volunteers grouped in two study arms, each consisting of 40 individuals, by including diabetic (T2D study arm) or PM (OP study arm) women wF or woF [60]. This review will focus exclusively on the OP study arm. Twenty-three miRs were found differentially expressed in this group by using low-density qPCR array. Subsequent multivariate classification analysis identified down-regulated (log_2_ FC < −1) miR-382-3p and up-regulated (log_2_ FC > 1) miR-188-3p in PM OP wF patients with respect to PM woF CTRs, although without reaching statistical significance, as the most promising biomarkers for OP.

Kocijan et al. investigated an miR serum profile correlated with low-traumatic fracture in a cohort of patients, including 10 PM OP and 26 idiopathic OP, comparing them with 39 age-, height-, and weight-matched woF CTRs. Sixteen of the 187 miRs analyzed were lower and three were higher in patient groups than in CTRs, and they were identified as significant discriminators of fragility fractures, by ROC analysis, with a AUC value from 0.81 to 0.96 [61].

Two years later, some of the co-authors of the previous study, partly replicating its experimental phases, verified that miR-550a-3p, miR-324-3p, and miR-29b-3p exhibited a significant correlation to bone architecture and dynamic histomorphometric parameters [62], enhancing the potential value of these miRs as bone biomarkers.

The levels of miR-124-3p and miR-2861 were also found up-regulated, and miR-21-5p, miR-23a, and miR-29a-3p were down-regulated, in serum samples of 35 low bone mass wF patients compared with 30 CTRs [50]. Moreover, miR-21-5p levels were also lower in OP wF than OP woF patients. (Further details have been reported in the previous Section 3.1.1).

Serum levels of 95 miRs were tested in pooled samples from 30 PM OP wF and from 30 age-matched CTR woF subjects [63]. The levels of five miRs (miR-125b, miR-30, miR-4665-3p, miR-5914, and miR-96) were validated as significantly increased in PM OP wF serum samples of the same cohort by qPCR methodology. However, only the levels of miR-125b, miR-30, and miR-5914 exhibited an adequate FC. Finally, as miR-125b was the most up-regulated and showed a higher AUC value, the authors proposed it as a suitable biomarker for OP.

Kelch S et al. investigated the serum levels of nine miRs, which were previously described as dysregulated in OP patients [64]. In this novel study [65], 14 OP wF (seven females and seven males) and 14 woF CTR (seven females and seven males) individuals were recruited. qPCR-based analysis showed that both miR-93-5p and miR-125b-5p were increased in the serum of OP females only, while miR-21-5p, miR-24-3p, miR-100-5p, miR-23a-3p, miR-122-5p, miR-1243p, and miR-148a-3p were increased in all OP patients compared to CTRs. In addition, the first five miRs were also up-regulated in human OBs derived from OP patients, while miR-21-5p, miR-93-5p, miR-100-5p, miR-122-5p, miR-1243p, miR-125b-5p, and miR-148a-3p were up-regulated in human OCs. Lastly, the expression of miR-21-5p, miR-23a-3p, miR-24-3p, miR-93-5p, miR-100-5p, and miR-148a-3p in hOBs were positively correlated with BMD. Overall, their results suggest that these miRs may be important for an earlier diagnosis of OP.

Mandourah et al. also identified the down-regulation of miR-122-5p in the serum of 18 OP wF when compared to 33 OP woF patients and 12 CTR subjects. Moreover, miR-4516 was also more decreased in plasma of low bone mass wF than low bone mass woF patients and CTRs [51]. (Further details have been reported in the previous Section 3.1.1).

Li et al. chose to examine the c-miR-133a potentiality as a biomarker in distinguishing PM OP patients with bone fractures to CTR woF subjects, as it was previously reported to be dysregulated and negatively correlated with BMD in PM OP [25]. They confirmed that the serum levels of miR-133a were higher in patients than in age-matched CTRs, and it had a negative correlation with lumbar spine BMD. Finally, functional in vitro and in vivo studies on rat models confirmed the role of miR-133a in inducing OC differentiation, promoting osteoporosis onset, by targeting the *RUNX2* gene [66]. Therefore, it was a valid candidate, not only as a potential biomarker for osteoporosis and for the risk of low traumatic fractures, but also as a therapeutic target for the treatment of this pathology.

Ramírez-Salazar et al. also detected higher levels of miR-23a-3p and miR-140-3p in the serum of 21 OP wF with respect to 22 CTRs by [52]. (Further details have been reported in the previous Section 3.1.1).

None of the three selected miRs (miR-26a-5p, miR-34a-5p, and miR-223-5p) by Pickering et al. [67] showed significant differences in the serum of PM OP wF patients when compared to woF PM CTRs.

Interestingly, seven serum miRs (miR-375, miR-532-5p, miR-19b-3p, miR-152-3p, miR-23a-3p, miR-335-5p, and miR-21-5p) were all up-regulated when the PM OP wF patients (*n* = 24) were compared with PM OP woF (*n* = 35) or CTRs (*n* = 40). Moreover, miR-19b-3p was positively correlated with the serum levels of three BTMs (osteocalcin, BALP, and CTx) [68].

Conversely, Sun et al. found reduced levels of miR-19b-3p by comparing 24 plasma samples from OP wF patients vs. CTRs [53]. (Further details were reported in the previous Section 3.1.1).

In Table 2, c-miRs tested as diagnostic biomarkers for OP and the assessment of fragility fracture risk are listed, indicating the protocols used in each study.

#### 3.2.2. OP wF vs. CTR wnOF

The first study that aimed to evaluate the potential role of cell-free c-miRs as biomarkers for OP diagnosis was conducted in 2014 by Seelinger et al. [64]. They initially screened 83 human miRs in two pooled serum samples from 10 wF OP patients and 10 wnOF CTR subjects, identifying 11 up-regulated miRs in the patient group. The subsequent validation stage performed in a larger set of samples (30 OP wF and 30 CTR wnOF), also including miR-93 and miR-637, which were known to be bone development associated, confirmed increased levels of nine miRs, namely miR-21-5p, miR-23a-3p, miR-24-3p, miR-100-5p, miR-122-5p, miR-124-3p, miR-125-5p, miR-148a-3p, and miR-93, in the serum of OP wF patients. Of these, miR-21-5p, miR-23a-3p, miR-24-3p, miR-100-5p, and miR-125-5p were also found up-regulated in bone tissue biopsies of OP patients. The diagnostic power of these nine miRs was confirmed by performing ROC analysis.

Ten bone metabolism-related miRs were assessed in a cohort of 60 individuals, 45 OP wF and 15 CTR wnOF, by using qPCR platform and *U6* as eRG for data normalization. Five of these miRs (miR-24-3p, miR-27a-3p, miR-100, miR-125b, and miR-122a) were found up-regulated in both serum and bone tissue samples of patients compared to CTR, whereas miR-144-3p were down-regulated in the same samples. Moreover, the levels of miR-128 and miR-145-5p were higher in bone and serum patients, respectively. Since miR-144-3p had not been OP correlated, the authors carried out further functional studies on CD14+ peripheral blood mononuclear cells, finding that this miR is able to affect hOCL differentiation via RANK binding, proliferation, and apoptosis. Together, these results indicate that miR-144-3p could be a useful marker for osteoporosis and a promising therapeutic candidate for the treatment of this disease [69].

Li et al. analyzed a panel of ten miRs known to be involved in bone metabolism on serum and bone tissue biopsies obtained from six OP wF and six CTR wnOF patients. Four miRs (miR-363-3p, miR-214, miR-103a, and miR-148a) were up-regulated, while another two miRs (miR-34a and miR-503) were down-regulated in both types of sample. Furthermore, miR-2816 and miR-3615 were down-regulated in serum and bone tissue samples, respectively. However, miR-363-3p showed higher FC value, so its involvement in bone metabolism was investigated. The functional study results showed that miR-363-3p, on the one hand, acts by promoting osteoclastogenesis, and on the other, it acts by inhibiting OB differentiation, by targeting the PTEN gene [70].

In a recent study, miR-483 was reported to be up-regulated in both serum and bone tissue samples from PM OP wF patients (*n* = 30) compared to those obtained by age-, BMI-matched normal BMD wnOF individuals (*n* = 36). In addition, with a protocol similar to that described above, the authors demonstrated that via binding the *IGF2* gene, this miR reduces osteoclastogenic marker expression in CD14+ cells, promoting OP development [71].

Table 3 reports c-miRs identified as biomarkers to distinguish OP patients wF from CTRs wnOF and related protocols.

## 4. Conclusions

OP is a multifactorial disabling disease that is characterized by decreasing bone mass, often followed by low-traumatic fracture occurrences with a strong negative impact on the quality of life and important economic repercussions. The availability of valid diagnostic tools to identify the onset, progression, and manifestation (i.e., fractures) of OP, as well as to evaluate the treatment administration efficacy, has allowed physicians to manage this pathology more effectively. However, these tools are not yet able to guarantee the necessary sensitivity and specificity. In recent years, the scientific community has focused its attention on a novel class of potential diagnostic biomarkers named circulating cell-free miRs. Despite the promising data, their clinical use is still under evaluation. Among the reasons that have limited their use is a lack of standardized protocols for their measurement, in addition to the absence of univocal data in the literature, which do not allow the establishment of specific guidelines.

While these issues are studied, our group decided to write this review by grouping the published papers on PubMed that concern this topic, based on their specific experimental design, in order to help to establish which could be the best miRs to use according to the research goals set or the clinical conditions of patients. For this purpose, the Venn diagrams depicted in Figure 1 and Figure 2 summarize all the c-miRs identified as possible biomarkers for OP diagnosis; furthermore, the c-miRs included in the diagram of Figure 2 could be also used for fragility fracture risk assessment. In particular, analyzing the diagram in Figure 1, it emerges that miR-208 could be the ideal candidate to use when the research goal is to distinguish OP patients woF from those with normal BMD. Similarly, miR-21-5p could be evaluated when the experimental design foresees the comparison between patients with LBMD and healthy controls. Finally, both the miR-122-5p and miR-30b-5p could be included in the list of miRs to be analyzed when the study population is composed by only Ope patients. Instead, Figure 2 suggests miR-93-5p as the most effective molecule in distinguishing OP patients wF from woF/wnOF controls, while the miR-30 and a wide group of miRs (miR-125b-3p, miR-124-3p, miR-148a-3p, miR-21-5p, miR-24-3p, miR-23a-3p, miR-2861, miR-100-5p, and miR-122-5p) could be evaluated when the CTRs are woF or wnOF subjects, respectively. Consequently, this miR could be taken into particular consideration when the aim of the research is to identify specific prognostic markers that, combined with the WHO-FRAX score, could increase the prediction of fragility fracture risk. Finally, Figure 3, which is a combination of the previous ones, shows the various c-miRs in the general OP context. This diagram allows the identification of four specific miR groups that are potentially usable for studies aimed to distinguish OP patients from CTRs woF (miR-190a-3p, miR-29a-3p, miR-29b-3p, miR-203a, miR-328a-3p, let-7g-5p, miR-96, and miR-30b), or from CTRs woF/wnOF (miR-144-3p and miR-125b-5p), or OP wF patients from normal BMD subjects (miR-93-5p and miR-24-3p). The last group of miRs (miR-2861, miR-122-5p, miR-125b-5p, miR-124, miR-100, miR-23a-3p, miR-148-3p, and miR-21-5p) could be evaluated for experimental designs that provide reduced inclusion criteria regarding the clinical characteristics of OP patients.

In conclusion, although further studies are needed to overcome the practical limitations that prevent the use of c-miRs as biomarkers for OP in clinical routine, the promising data described in the literature indicate that these small molecules possess the characteristics to be used as diagnostic and prognostic tools for this pathology.

## Figures and Tables

**Figure 1 ijms-21-06927-f001:**
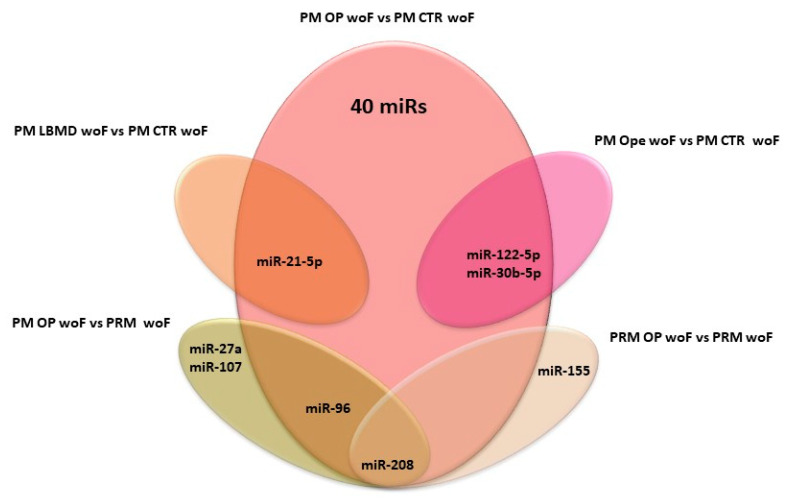
Venn diagram summarizes all the miRs identified as possible biomarkers for LBMD. PM: post-menopause; OP: osteoporosis; PRM: pre-menopause; LBMD: low bone mass density; CTR: control; woF: without fracture.

**Figure 2 ijms-21-06927-f002:**
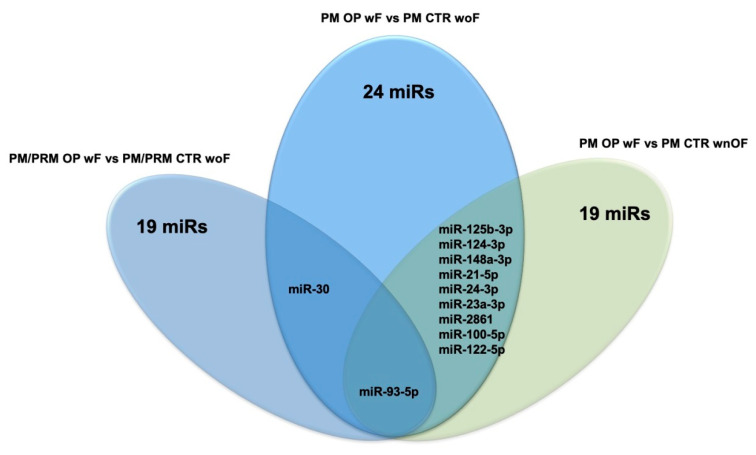
Venn diagram reports the potential biomarker miRs for fragility fracture risk. PM: post-menopause; OP: osteoporosis; PRM: pre-menopause; CTR: control; woF: without fracture; wF: with fracture; wnOF: with no-osteoporosis fracture.

**Figure 3 ijms-21-06927-f003:**
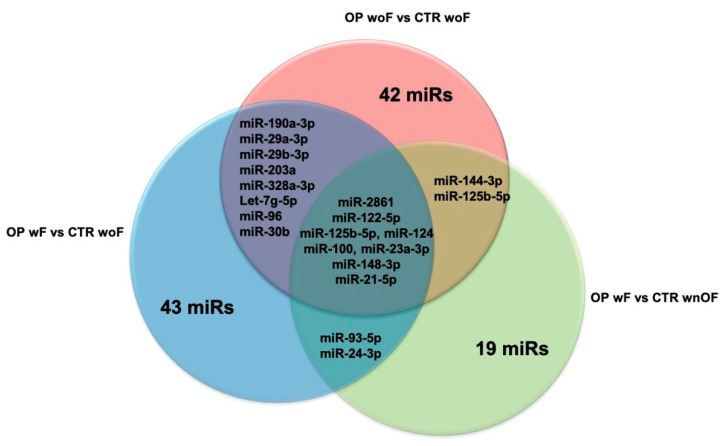
Venn diagram shows all the miRs in the general OP context. OP: osteoporosis; woF: without fracture; wF: with fracture; wnOF: with no-osteoporosis fracture.

**Table 1 ijms-21-06927-t001:** Overview of studies that have identified cell-free microRNAs (c-miRs) as biomarkers to distinguish osteoporosis/osteopenia (OP/Ope) patients from normal bone mineral density (BMD) without fragility fracture (woF) subjects.

Study Design	OP Diagnosis	c-miR Source	Platforms for c-miR Quantification		Validate c-miR Biomarkers	Normalization Strategy	AUC Value	c-miR Correlations	Ref
			Prescreening	Validation					
PM: OP (40), Ope (40), CTR (40)	DXA	Plasma	/	qPCR	miR-21-5p (↓), miR-133a (↑), in OP and Ope vs. CTR	miR-16	/	miR-21-5p and miR-133a/h- and BMD (+) and (−) respectively	[25]
PM: OP (10), Ope (7) CTR (19)	DXA	Serum	Microarray	qPCR	miR-30-5p (↓) in OP and Ope vs. CTR; miR-328a-3p, miR-103-3p and miR-142-3p (↓) in OP vs. CTR	miR-25-3p	miR-30-5p: 0.79; miR-328a-3p: 0.87; miR-103-3p: 0.8; miR-142-3p: 0.79	All four miRs/BMD (+)	[26]
PM: OP (17), CTR (57)	DXA FRAX	Plasma	/	qPCR	miR-148-3p (↑) in OP vs. CTR	let-7a-5p and miR-16-5p	/	miR-126-3p/BMD (+); miR-423-5p/FRAX (−)	[27]
PM OP (20), PRM CTR (20), PM CTR (20)	DXA	Serum	Microarray	qPCR	miR-96 (↑) in OP vs. PM CTR and (↓) in OP vs. PRM CTR	U6	/	miR-96/BALP (−)	[28]
OP (9), CTR (9)	DXA	Serum	Microarray	qPCR	miR-10b-3p and miR-100 (↑), and miR-328-3p and let-7g-5p (↓) in OP vs. CTR	U6	miR-10b-3p: 0.87	/	[29]
OP (60), CTR (60)/OP (60), CTR (40)	QCT	Serum	/	qPCR	miR-203 (↓) in OP vs. CTR	/	/	/	[30,31]
PM: LBMD (109), CTR (41)	DXA	Plasma	/	qPCR	miR-21-5p (↑) in LBMD vs. CTR	cel-miR-39	/	miR-21-5p/BMD (−)	[32]
PM: LBMD (46), CTR (13)	DXA	Serum	/	qPCR	miR-21-5p, miR-23a-3p (↑), and miR- 125b-5p (↓) in LBMD vs. CTR	miR-16-5p, miR-93-5p, miR-191-5p	/	miR-21-5p and miR-133a-3p/BMD (−) and (+), respectively; miR-23a-3p/TRAP5b (+)	[33]
OP (48), CTR (48)	/	Serum	/	qPCR	miR-17, miR-20a, miR-21 and miR-29a (↓) in OP vs. CTR	U6	/	/	[34]
OP (/), CTR (/)	/	Serum	/	qPCR	miR-579-3p (↑) in OP vs. CTR	U6	/	/	[35]
PM: OP (20), CTR (20)	μCT	Serum	/	qPCR	miR-28, miR-101 and miR-373 (↓) in OP vs. CTR	U6	/	/	[36]
OP (/), CTR (/)	/	Serum	/	qPCR	miR-30a-5p (↑) in OP vs. CTR	/	/	/	[37]
PM: OP (26), CTR (29)	/	Serum	/	qPCR	miR-410 (↑) in OP vs. CTR	U6	/	/	[38]
PM: OP (15), CTR (15)	/	Serum	/	qPCR	miR-144 (↑) in OP vs. CTR	U6	/	miR-144/Sfrp1 (+)	[39]
OP (30), CTR (30)	/	Serum	/	qPCR	miR-200a-3p (↑) in OP vs. CTR	U6	/	/	[40]
PM: OP (15), CTR (15)	/	Serum	/	qPCR	miR-338-3p and miR-3065-5p (↑) in OP vs. CTR	U6	miR-338-3p: 0.74; miR-3065-5p: 0.87	/	[41]
OP (42), CTR (42)	/	Serum	/	qPCR	miR-19a-3p (↓) in OP vs. CTR	U6	/	/	[42]
OP (60), CTR (60)	DXA	Serum	/	qPCR	miR-21-5p (↑) in OP vs. CTR	/	/	miR-21-5p/RUNKL and OPG (+)	[43]
OP (120), CTR (120)	/	Plasma	/	qPCR	miR-100 (↑) in OP vs. CTR	/	miR-100: 0.89	miR-100/BMD (−)	[44]
OP (30), CTR (30)	/	Serum	/	qPCR	miR-375 (↑) in OP vs. CTR	U6	/	/	[45]
OP (30), CTR (30)	/	Serum	/	qPCR	miR-429 (↑) in OP vs. CTR	/	/	/	[46]
OP (/), CTR (/)	/	Serum	/	qPCR	miR-1286 (↑) in OP vs. CTR	/	/	/	[47]
OP (/), CTR (/)	/	Serum	/	qPCR	miR-217 (↑) in OP vs. CTR	/	/	/	[48]
PM+PRM OP (70), PM+PRM CTR (70)	DXA	Serum	/	qPCR	miR-208a-3p (↑) in OP vs. CTR; miR-155-5p and miR-637 (↑) in PM OP vs. PM CTR; miR-155-5p (↓) PRM OP vs. PRM CTR	SNORD68	PM OP vs. PM CTR: miR-208a-3p: 0.82; miR-155-5p: 0.9; miR-637: 0.63. PM OP vs. PM CTR: miR-208a-3p: 0.85; miR-155-5p: 0.83; miR-637: 0.81.	All 3 miRs/BMD (−)	[49]
PM: LBMD (35), CTR (30)	DXA	Serum	/	qPCR	miR-124 and miR-2861 (↑) LBMD vs. CTR; miR-21-5p, miR-29a-3p, miR-23a-3p (↓) in LBMD vs. CTR	/	miR-21-5p: 0.66; miR-29a-3p: 0.61; miR23a-3p: 0.63	/	[50]
OP (33), Ope (61), CTR (12)	DXA	Serum and plasma	qPCR array	qPCR	miR-122-5p (↓ in serum) OP vs. Ope, OP vs. N, Ope vs. N; miR-4516 (↓ in plasma) OP vs. N	SNORD96A, RNU6–6P	miR-122-5p: 0.67; miR-4516: 0.73; miR-122-5p+miR-4516: 0.75	miR-122-5p and miR-4516/BMD (+)	[51]
PM OP (16), PM Ope (28), PM CTR (22)	DXA or fracture	Serum	miRtaqman array card	qPCR	miR-23b-3p and miR-140-3p (↑) in OP/ Ope vs. CTR; miR-885-5p (↑) in Ope vs. CTR	RNU6-1	miR-140-3p: 0.96 for OP, 0.84 for Ope; miR-23b-3p: 0.69 for OP, 0.73 for Ope; miR-885-5p: 0.69 for Ope	miR-23b-3p and miR-140-3p/BMD (−)	[52]
OP (24), CTR (24)	DXA	Plasma	Microarray	qPCR	miR-19b-3p (↓) in OP vs. CTR	U6	miR-19b-3p: 0.93 OP,	miR-19b-3p/BMD (+)	[53]
PM OP (81), PRM CTR (74)	DXA	Serum	Microarray	qPCR	miR-27a (↓) in OP vs. CTR	U6	/	/	[55]
PM: OP (17), Ope (14), PRM CTR (14)	DXA	Serum	Microarray	qPCR	miR-204-3p (↑) in OP/Ope vs. CTR; miR-181c-5p and miR-497-5p (↓) in OP/Ope vs. CTR	5S	miR-204-3p: 0.96, miR-181c-5p: 0.87 and miR-497-5p: 0.92 for OP; miR-204-3p: 0.77, miR-181c-5p: 0.69 and miR-497-5p: 0.75 for Ope;	miR-204-3p/BMD (−), BMI and CTx (+); miR-181c-5p and miR-497-5p/BMD (+)	[56]

↑: up-regulated, ↓: down-regulated, +: positive correlation, −: negative correlation.

**Table 2 ijms-21-06927-t002:** Overview of studies that evaluated c-miRs to distinguish OP/Ope patients with fragility fracture (wF) from normal bone mass controls (CTRs) woF.

Study Design	OP Diagnosis	c-miRs Source	Platforms for c-miRs Quantification		Validate c-miR Biomarkers	Normalization Strategy	AUC Value	c-miR Correlations	Ref
			Prescreening	Validation					
OP F (18), OP (33), Ope F (15), CTR (12)	DXA	Serum and plasma	qPCR array	qPCR	miR-122-5p (↓ in serum) OP F vs. OP/CTR; miR-4516 (↓ in plasma) LBMD F vs. LBMD	SNORD96A, RNU6–6P	miR-122-5p: 0.67; miR-4516: 0.73; miR-122-5p+miR-4516: 0.75	miR-122-5p and miR-4516/BMD (+)	[51]
OP F (21), PM CTR (22)	DXA or fracture	Serum	miRtaqman array card	qPCR	miR-23b-3p and miR-140-3p (↑) in OP F vs. CTR;	RNU6-1	/	miR-23b-3p and miR-140-3p/BMD (−)	[52]
OP F (24), CTR (24)	DXA	Plasma	Microarray	qPCR	miR-19b-3p (↓) in OP F vs. CTR	U6	miR-19b-3p: 0.93 OP, 0.95 OP F	miR-19b-3p/BMD (+)	[53]
PM: OP F (19), CTR (18)	Fracture	Serum	qPCR array	qPCR	let-7g-5p, miR-328-3p, miR-22-3p (↓) in OP F vs. CTR	/	/	/	[58]
PM: OP F (15), OA CTR (12)	DXA	Serum	qPCR array	qPCR	miR-21-5p (↑) in OP F vs. CTR	miR-140-3p, miR-93-5p	miR-21-5p: 0.87	miR-21-5p/CTx and OCN (+)	[59]
PM: OP F (20), CTR (20)	DXA	Serum	/	low density qPCR array	miR-550a-5p, miR-203a and miR-330-3p (↑) and miR-328-3p (↓) in OP F vs. CTR	/	/	/	[60]
PM OP F (10), IOP F (26), PM CTR (11), PMR/male CTR (28)	/	Serum	/	qPCR array	miR-152-3p, miR- 335-5p and miR-320a (↑), and let-7b-5p, miR-7-5p, miR- 16-5p, miR-19a-3p, miR-19b-3p, miR-29b-3p, miR-30e-5p, miR- 93-5p, miR-140-5p, miR-215-5p, miR-186-5p, miR-324-3p, miR-365a-3p, miR-378a-5p, miR-532-5p, and miR-550a-3p (↓) in fractured group vs. CTR	miRs Ct average	miR-152-3p: 0.96, miR-30e-5p: 0.96, miR-324-3p: 0.95, miR-140-5p: 0.95, miR-19b-3p: 0.94, miR-335-5p: 0.94, miR-19a-3p: 0.93, miR-550a-3p: 0.91, miR-186-5p: 0.9, miR-532-5p: 0.9, miR-378a-5p: 0.87, miR-320a: 0.87, miR-93-5p: 0.88, miR-16-5p: 0.86, miR-215-5p: 0.85, let-7b-5p: 0.85, miR-7-5p: 0.82, miR-29b-3p: 0.84, and miR- 365a-3p: 0.81 for fracture group vs. CTR	miR-140-5p/BMI (−); miR-320a/BMI (+); miR-29b-3p/PIPN (+); miR-365a-3p/PTH, TRAP5, PIPN, OCN (+); miR19a-3p, miR-324-3p, miR-532-5p and miR-93-5p/BMD (+)	[61]
PM: OP F (30), CTR (30)	Fracture	Serum	Microarray	qPCR	miR-30, miR-96, miR-125b, miR-5914, miR-4665-3p (↑) in OP vs. CTR	U6	miR-125b: 0.9; miR-30: 0.76; miR-5914: 0.7	/	[63]
PM: OP F (14), CTR (14)	DXA	Serum	/	qPCR	miR-21-5p, miR-24-3p, miR-23a-3p, miR-122-5p, miR-124-3p, miR-148a-3p and miR-100-5p (↑) in OP F vs. CTR; miR-93-5p and miR-125b-5p (↑) in OP female vs. CTR	SNORD96a	/	/	[65]
PM OP F (10), CTR (10)	DXA	Serum	/	qPCR	miR-133a (↑) in OP F vs. CTR	U6	/	miR-133a/BMD (−)	[66]
PM: LBMD (35), LBMD F (24), CTR (40)	DXA	Serum	/	qPCR	miR-375, miR-532-5p, miR-19b-3p, miR-152-3p, miR-23a-3p, miR-335-5p, and miR-21-5p (↑) in LBMD F vs. LBMD and CTR	miR-451a	/	miR-19b-3p/OCN, BALP and CTx (+)	[68]

↑: up-regulated, ↓: down-regulated, +: positive correlation, −: negative correlation, F: with fracture.

**Table 3 ijms-21-06927-t003:** Overview of studies that identified c-miRs as biomarkers to distinguish OP patients wF from CTRs wnOF.

Study Design	OP Diagnosis	c-miRs Source	Platforms for c-miR Quantification		Validate c-miR Biomarkers	Normalization Strategy	AUC Value	c-miR Correlations	Ref
			Prescreening	Validation					
OP F (40), CTR wnOF (40)	DXA	Serum	qPCR array	qPCR	miR-21-5p, miR-23a-3p, miR-24-3p, miR-100-5p, miR-122-5p, miR-124-3p, miR-125-5p, miR-148a-3p and miR-93 (↑) in OP F vs. CTR	SNORD96a and RNU6	miR-21-5p: 0.63, miR-23a-3p: 0.63, miR-24-3p: 0.63, miR-100-5p: 0.69, miR-122a-5p: 0.77, miR-124a-3p: 0.69, miR-125b-5p: 0.76, miR-148a-3p and miR-93: 0.68	/	[64]
OP F (45), wnOF CTR (15)	/	Serum	/	qPCR	miR-24-3p, miR-27a-3p, miR-100, miR-125b, miR-122a and miR-145-5p (↑) in OP F vs. CTR	U6	/	/	[69]
OP F (6), wnOF CTR (6)	/	Serum	/	qPCR	miR-363-3p, miR-214, miR-103a, miR-148a (↑), and miR-2861, miR-34a, miR-503 (↓) in OP F vs. CTR	/	/	/	[70]
PM: OP F (30), wnOF CTR (36)	DXA	Serum	/	qPCR	miR-483 (↑) in OP F vs. CTR	U6	/	/	[71]

↑: up-regulated, ↓: down-regulated, F: with fracture.

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
