# Peer review of "Circulating MicroRNAs as Novel Biomarkers for Osteoporosis and Fragility Fracture Risk: Is There a Use in Assessment Risk?"

_ijms, 2020, doi:10.3390/ijms21186927_

Round 1

Reviewer 1 Report

This review paper, written by Simone Ciuffi et al., summarized circulating non-coding microRNA candidates that could be used as novel diagnostic markers for osteoporosis and fragile fracture risk. The authors investigated the circulation miRNA extensively by organizing experimental groups in a specific way, depending on the disease condition of patients with skeletal disorders. The authors then suggested that the miRNAs they have summarized would be useful in selecting good markers for osteoporosis research and diagnosis. In this report, each case of miRNA associated with osteoporosis is well organized according to the condition of each patient. Given the recent interest in circulating miRNAs (c-miRNAs) in the fields related to the OP and MSC biology, this review is timely and would be of value to readers in the related field. However, there are some points that should be explained further to support the conclusions enclosed.

  1. In the introduction of the manuscript, the authors described the history of miRNA and how c-miRNAs were first discovered. I think readers would have more questions about the history of miRNA, for example, about the fundamental reasons to use miRNAs as diagnostic markers. One of them may be the reason why a certain non-coding RNA is found in specific disease conditions, such as OP. Some non-coding RNAs can be created by DNA damage under specific environments related with certain diseases. Therefore, I’d like to recommend that authors should add to the introduction some contents regarding the fundamental reasons why miRNAs can be up- or down-regulated under a certain disease condition.
  2. The origin of c-miRNA is known to be difficult to trace; c-miRNA can be derived from dead blood cells, as well as secreted from living cells. C-miRNAs are found in a form combined with AGO proteins or exosomes. For this reason, the c-miRNA is stable because exosomes are impermeable to RNase, as well as AGO proteins binding to c-miRNA prevent enzymatic disruption, so c-miRNA can be an ideal candidate as a new diagnostic biomarker for various diseases. The above findings may be one reason why c-miRNA is worthy of being used as a diagnostic marker, but the authors briefly summarize the cases where c-miRNA has been studied and did not review the biologically important reasons for c-miRNA. Therefore, the introduction of this paper will require a review of the molecular biological value of c-miRNA.
  3. Table 1: It is difficult to compare the potential diagnostic markers for osteoporosis by Table 1, which the authors have organized. The authors compared the quantitative differences of c-miRNAs through comparisons by each group (PM OP/Ope vs PM CTR; PM OP vs PRM CTR; OP wF vs CTR woF; OP wF vs CTR wnOF) in the text of the manuscript, but Table 1 was not summarized by comparison between groups, but all the contents they investigated are summarized. The authors have made Table 1 more convenient for readers to compare c-miRNAs between groups. Please subdivide Table 1.
  4. Line 103-113: Authors summarized that miR-21 and miR-133a could be one of potential diagnostic marker for OP and Ope patients. According to the authors, miR-21 and miR-133a are shown to be only different expression patterns in PM OP/Ope patients compared to PM CTR. These differences in expression alone cannot be considered as diagnostic markers for osteoporosis; further explanations of their precise molecular biological mechanisms are likely to be needed. In addition, it will be necessary to describe the differences between these two miRNAs in other diseases. This is because we need to know if miR-21 and miR-133a can be specific diagnostic markers for osteoporosis.
  5. Line 267-270: The explanation is too short. Put together the things that Reference 48 means. I think the authors of review papers should write review papers, including a sufficient interpretation of the papers they cited.
  6. Conclusion: Discussion and conclusion are too short. What are the highly likely circulating miRNAs as biological markers for the diagnosis of osteoporosis, both experimentally and theoretically? And do authors think that this review manuscript provides the appropriate guidelines for selection of circulating miRNAs to use for the diagnosis of osteoporosis? In-depth discussion of the aforementioned contents, including appropriate analysis and interpretation, should be included in this manuscript.
  7. There are many English grammatical errors in this manuscript.

Author Response

  1. In the introduction of the manuscript, the authors described the history of miRNA and how c-miRNAs were first discovered. I think readers would have more questions about the history of miRNA, for example, about the fundamental reasons to use miRNAs as diagnostic markers. One of them may be the reason why a certain non-coding RNA is found in specific disease conditions, such as OP. Some non-coding RNAs can be created by DNA damage under specific environments related with certain diseases. Therefore, I’d like to recommend that authors should add to the introduction some contents regarding the fundamental reasons why miRNAs can be up- or down-regulated under a certain disease condition.

RESPONSE 1: As recommended, we have expanded the description of the c-miRs features that make them efficient biomarkers. (LINE 75-83; “Moreover, their elevated stability in biological fluids has been shown, and imputable to encapsulation in membrane-bound vesicles, such as shedding microvesicles, exosomes, and apoptotic bodies [20], or to their association with proteins, such as Argonaute 2, high-density lipoproteins, and nucleophosmin 1 [21]. Finally, these molecules possess the three fundamental features to be used as biomarkers: 1. they are easily obtainable (with minimal or non-invasive procedures), and detectable in reproducible manner with high specificity and sensitivity (via various platforms: real-time qRT-PCR, microarray, and NGS); 2. they can add new information with regards to disease of interest; 3. they help the physicians’ decision-making [22]. These features make c-miRs candidates for novel potential diagnostic biomarkers”).

  1. The origin of c-miRNA is known to be difficult to trace; c-miRNA can be derived from dead blood cells, as well as secreted from living cells. C-miRNAs are found in a form combined with AGO proteins or exosomes. For this reason, the c-miRNA is stable because exosomes are impermeable to RNase, as well as AGO proteins binding to c-miRNA prevent enzymatic disruption, so c-miRNA can be an ideal candidate as a new diagnostic biomarker for various diseases. The above findings may be one reason why c-miRNA is worthy of being used as a diagnostic marker, but the authors briefly summarize the cases where c-miRNA has been studied and did not review the biologically important reasons for c-miRNA. Therefore, the introduction of this paper will require a review of the molecular biological value of c-miRNA.

RESPONSE 2: The answer to this note is integrated in the previous answer.

  1. Table 1: It is difficult to compare the potential diagnostic markers for osteoporosis by Table 1, which the authors have organized. The authors compared the quantitative differences of c-miRNAs through comparisons by each group (PM OP/Ope vs PM CTR; PM OP vs PRM CTR; OP wF vs CTR woF; OP wF vs CTR wnOF) in the text of the manuscript, but Table 1 was not summarized by comparison between groups, but all the contents they investigated are summarized. The authors have made Table 1 more convenient for readers to compare c-miRNAs between groups. Please subdivide Table 1.

RESPONSE 3: As required, we have subdivided Table 1 into three new tables, based on the three different research experimental designs (LINE 351-357, Table 1; LINE 435-440, Table 2; LINE 475-480, Table 3)

  1. Line 103-113: Authors summarized that miR-21 and miR-133a could be one of potential diagnostic marker for OP and Ope patients. According to the authors, miR-21 and miR-133a are shown to be only different expression patterns in PM OP/Ope patients compared to PM CTR. These differences in expression alone cannot be considered as diagnostic markers for osteoporosis; further explanations of their precise molecular biological mechanisms are likely to be needed. In addition, it will be necessary to describe the differences between these two miRNAs in other diseases. This is because we need to know if miR-21 and miR-133a can be specific diagnostic markers for osteoporosis.

RESPONSE 4: Based on the data reported by authors, the plasma levels of both miRs were also correlated with hip and spine BMD values (LINE 127-129), which corroborates the potential role of these miRs as biomarkers for OP diagnosis. Moreover, in the introduction of their paper, the authors cited various studies that demonstrated the role of these two miRs in bone metabolism. The authors also cited the Seelinger’s paper, in which it was reported that circulating miR-21 was found dysregulated in OP serum patients. Subsequent, other authors have demonstrated the potential role of miR-21 (i.e. Yavropoulou M, 2017; Zhao Z, 2018) and miR-133a (i.e. Li Z, 2018; Pala E, 2019) as potential biomarkers for OP diagnosis. In conclusion, based on aforementioned contents, we believe correct what the authors reported in their paper, namely that these molecules could be used as potential biomarkers for OP diagnosis.

  1. Line 267-270: The explanation is too short. Put together the things that Reference 48 means. I think the authors of review papers should write review papers, including a sufficient interpretation of the papers they cited.

RESPONSE 5: As required, we have expanded the description of results regarding Reference 48. (LINE 284-291; “Yavropoulou M et al. investigated levels of twelve bone-related miRs and two miRs reported as dysregulated in monocytes of OP patients. The serum samples were obtained from a cohort of LBMD PM patients (35 wF and 35 woF) and PM CTRs (30). By using qPCR assay, and SNORD95, SNORD96A and RNU6-2 genes as eRG, the authors identified 2 up-regulated miRs (miR-124-3p and miR-2861) and 3 down-regulated (miR-21-5p, miR-23a, and miR-29a-3p) in patients, when compared to CTRs. Moreover, miR-21-5p was also found down-regulated when its serum levels in fractured patients were compared with OP woF. These results suggest that these five c-miRs could be employed as biomarkers to distinguish LBMD patients from CTRs subjects [50]”). Highlighted in yellow the new parts added

  1. Conclusion: Discussion and conclusion are too short. What are the highly likely circulating miRNAs as biological markers for the diagnosis of osteoporosis, both experimentally and theoretically? And do authors think that this review manuscript provides the appropriate guidelines for selection of circulating miRNAs to use for the diagnosis of osteoporosis? In-depth discussion of the aforementioned contents, including appropriate analysis and interpretation, should be included in this manuscript.

RESPONSE 6: As recommended, we have expanded the paragraph of conclusions, indicating the circulating miRs most represented in each of the three different experimental designs (based on the analysis of Venn graphs, Fig 1-3), and which could therefore be included in future studies, as potentially very effective as diagnostic biomarkers. (LINE 498-522; “In particular, analyzing the diagram in figure 1, it emerges that miR-208 could be the ideal candidate to use when the research goal is to distinguish OP patients woF from those with normal BMD. Similarly, miR-21-5p could be evaluated when the experimental design foresees the comparison between patients with LBMD and healthy controls. Finally, both the miR-122-5p and miR-30b-5p could be included in the list of miRs to be analyzed when the study population is composed by only Ope patients. Instead, figure 2 suggests miR-93-5p as the most effective molecule in distinguishing OP patients wF from woF/wnOF controls, while the miR-30 and a wide group of miRs (miR-125b-3p, miR-124-3p, miR-148a-3p, miR-21-5p, miR-24-3p, miR-23a-3p, miR-2861, miR-100-5p, and miR-122-5p) could be evaluated when the CTRs are woF or wnOF subjects, respectively. Consequently, this miR could be taken into particular consideration when the aim of the research is to identify specific prognostic markers that, combined with the WHO-FRAX score, could increase the prediction of fragility fracture risk. Finally, figure 3, which is a combination of the previous ones, shows the various c-miRs in the general OP context. This diagram allows the identification of four specific miR groups, potentially usable for studies aimed to distinguish OP patients from CTRs woF (miR-190a-3p, miR-29a-3p, miR-29b-3p, miR-203a, miR-328a-3p, let-7g-5p, miR-96, and miR-30b), or from CTRs woF/wnOF (miR-144-3p and miR-125b-5p), or OP wF patients from normal BMD subjects (miR-93-5p and miR-24-3p). The last group of miRs (miR-2861, miR-122-5p, miR-125b-5p, miR-124, miR-100, miR-23a-3p, miR-148-3p and miR-21-5p) could be evaluated for experimental designs that provide reduced inclusion criteria regarding the clinical characteristics of OP patients.

In conclusion, although further studies are needed to overcome the practical limitations that prevent the use of c-miRs as biomarkers for OP in clinical routine, the promising data described in the literature indicate that these small molecules possess the characteristics to be used as diagnostic and prognostic tools for this pathology.”)

  1. There are many English grammatical errors in this manuscript.

RESPONSE 7: We have correct the English grammatical errors

Reviewer 2 Report

The manuscript entitles "Circulating microRNAs as novel biomarker for osteoporosis and fragility fractur risk: which one to choose for an accurate diagnosis" by Simone Ciuffi et al. is a very interesting review aiming to give a current and comprehensive update about circulating mRNAs as markers of osteporosis. This review article is well written and convincing. Authors underlining advantages and disavantages as well as their usefulness in fracture risk assessment.
Minor points:
-Sentences regarding animal model studies should be removed.
-Authors should provide a comment on the prognostic power of such markers.

Author Response

  1. Sentences regarding animal model studies should be removed.

RESPONSE 1: The sentences regarding animal model studies have been reported in this review because supporting the obtained data on the human model, they give greater robustness to the data themselves, without altering the meaning and / or purpose of the manuscript. Therefore, we believe they can be maintained in the text of the review.

  1. Authors should provide a comment on the prognostic power of such markers.

RESPONSE 2: We have expanded the discussion paragraph by adding a comment on the prognostic power of c-miRs, as requested (LINE 505-511; “Instead, figure 2 suggests miR-93-5p as the most effective molecule in distinguishing OP patients wF from woF/wnOF controls, while the miR-30 and a wide group of miRs (miR-125b-3p, miR-124-3p, miR-148a-3p, miR-21-5p, miR-24-3p, miR-23a-3p, miR-2861, miR-100-5p, and miR-122-5p) could be evaluated when the CTRs are woF or wnOF subjects, respectively. Consequently, this miR could be taken into particular consideration when the aim of the research is to identify specific prognostic markers that, combined with the WHO-FRAX score, could increase the prediction of fragility fracture risk”)

Round 2

Reviewer 1 Report

The authors addressed all my previous concerns, and the manuscript is now greatly improved.

Thank you for your efforts.